# Wireless in-body sensing through genetically engineered bacteria

Ahmet Bilir[1], Merve Yavuz [2], Urartu Ozgur Safak Seker [3,4] ✉ &
Sema Dumanli [1] ✉

This paper introduces a class of wireless implantable sensors that integrate genetically engineered cells capable of detecting specific molecules for continuous monitoring. While synthetic biology enables cells to sense molecular targets, wireless communication of this information remains a challenge. Electromagnetic (EM) waves at cellular-scale wavelengths are strongly attenuated in tissue, necessitating centimeter-scale wavelengths for in-body links. Aligning cellular responses with these longer EM wavelengths enables effective interaction. In this work, the response of *Escherichia coli* is harnessed to trigger the controlled degradation of a passive microwave antenna, which is then monitored via backscatter communication. This approach converts cellular activity into detectable EM signals, eliminating the need for batteries or circuits. We demonstrate a wireless link between a passive, cell-based sensor in a human body phantom and an external receiver, achieving molecular-level sensing at 25 mm implant depth. Future implementations could couple bacterial responses to diverse molecular targets.

In the coming years, the rapid increase in the population, and specifically the elderly, will lead to a much higher demand for healthcare services that cannot be met with the existing system. The solution to this problem is only possible through a revolution that will change the way we receive healthcare. For this technological revolution in healthcare to take place, it is necessary not only to transform our hospitals and homes but also the way we monitor our bodies. Constantly monitoring our well-being with various implant sensors and detecting health issues before symptoms arise constitutes a significant step in this technological revolution[1-3].

Taking a closer look at the sensors in the literature, we observe that various implantable devices are used for monitoring, diagnosis, or treatment purposes[4,5]. Monitoring devices target applications, such as capsule endoscopy[6], brain-computer interfaces[7], glucose[8,9], pH[10], and intravascular pressure monitoring[11]. These devices primarily monitor physical parameters. For instance, the electromagnetic wave-based glucose detection system proposed in[12] detects changes in the dielectric constant of tissue, which is a secondary effect of glucose concentration, rather than conducting molecule-specific detection. With the existing implant sensor technology, it is not possible to monitor specific molecules, such as disease-specific biomarkers, which may be needed for early diagnosis in vivo in real time[13]. To address this challenge, we propose utilizing engineered living cells. This approach requires a specific recognition tool for the molecule of interest, but current electronic or optical settings cannot provide such instrumentation. However, biological systems already possess highly advanced sensor systems capable of recognizing any type of molecule within a biological process. Cells, therefore, can be exploited as advanced sensor systems, but it is important to recognize that these systems have evolved within a specific biological context. Thus, if cellular sensors are desired, they need to be reprogrammed for our specific needs. Synthetic biology offers a set of engineering tools and approaches for such programming[14,15].

Whole-cell molecular-level detection has the potential to open doors for high-precision diagnostics[16]. While electrochemical or nano-based biosensors with molecular detection capabilities are found in

[1]Electrical and Electronics Engineering Dept., Bogazici University, Istanbul, Turkey. [2]Aziz Sancar Research Center, Turkiye Biotechnology Institute, Health Institutes of Turkiye (TUSEB), Ankara, Turkey. [3]Synbiotik Biotechnology and Biomedical Technology, Ankara, Turkey. [4]UNAM- Institute of Materials Science and Engineering, Ankara, Turkey. ✉e-mail: urartu@bilkent.edu.tr; sema.dumanli@bogazici.edu.tr

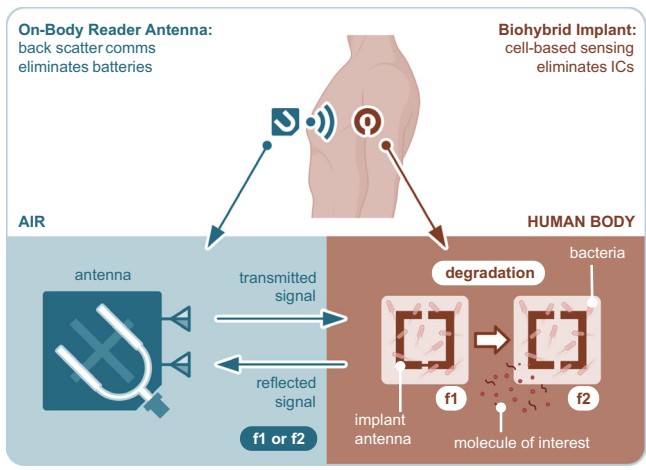

**Fig. 1 | The AntennAlive concept illustrating the principle of operation for genetically modified bacteria-based wireless sensing[30].** The illustration of the implant antenna was created in BioRender.com. *Bilir, A. (2025)* https://BioRender.com/rnfd7kj.

the literature[17–19], these sensors are in vitro and cannot provide real-time in vivo monitoring. More recent studies have started to address this limitation by developing platforms for real-time in vivo molecular sensing. For example, Saunders et al. introduces a set of generalizable molecular switch designs based on aptamers and antibodies for continuous in vivo detection. These sensors employ optical methods to convert molecular binding events into measurable signals, but they require a fiber-optic connection for signal acquisition and monitoring[20]. Similarly, Chen et al. demonstrates an optical sensing system capable of detecting specific molecules in vivo when implanted in the blood vessels of freely moving rats for up to one week[21]. Moutsiopoulou et al. also explores aptamer-based molecular sensing using optical moving techniques[22].

Chien et al. report an aptamer-based sensor for detecting kanamycin in freely moving rats, with data transmitted wirelessly via Bluetooth. However, this system requires battery and signal processing electronics[23]. Chen et al. discusses an implantable aptamer-based sensor that is inductively coupled to an external reader. Although it allows for wireless power and data transmission, the implant still contains integrated electronics for signal processing[24].

This paper presents a self-sustaining, electrically passive, implantable biosensor capable of molecular-level detection. The advent of such implantable biosensors is set to change medical diagnostics and monitoring.

Synthetic biology has been revolutionizing healthcare with its capability to reprogram living cells to function in a desired way[25,26]. For example, living cells–whether bacterial or mammalian–can be reprogrammed to operate as sensors[27,28]. Cells naturally detect and respond to stimuli, and synthetic biology allows us to manipulate their genetic circuitry, enabling the creation of transgenic sensors that respond to specific molecular stimuli in a controlled way[28,29]. In the context of cell-based sensing, the main challenge for electronics engineers lies in transferring the cells' responses–specifically the collected data–from reprogrammed cells to existing communication systems. In other words, we must convert cellular responses into signals that humans can interpret. This paper elaborates on our approach to achieving this with genetically modified bacteria. It is important to note that AntennAlive, the communication method employed, has international protection[30].

The principle of operation for genetically modified bacteria-based wireless sensing is illustrated in Fig. 1. The sensing is done using a bio-hybrid implant, which is wirelessly connected to an on-body reader antenna. The bio-hybrid implant is composed of a passive implant

antenna which is used as a reflector and a colony of genetically modified bacteria. The implant antenna, made of magnesium, is designed to degrade in a specific manner, such that as the degradation proceeds, its resonant frequency follows a predetermined pattern. This biodegradation is tracked by an on-body reader antenna based on backscatter communication. If the bacteria are genetically modified to increase the degradation speed in the presence of a molecule of interest, the operation will be complete. There are various biodegradable implant sensors proposed in the literature that explore similar wireless connections as proposed here[31,32]. In this study, we use *Escherichia coli* (E. coli) that includes a genetic circuit to accelerate the degradation of magnesium. We compare two cases: one where this genetic circuit is disabled and one where it is enabled. The molecule of interest acts as the enabler, though it is beyond the scope of this paper. We focus on demonstrating how to form a wireless link with genetically modified bacteria and present the proof of concept for the communication method.

To better understand the role of genetic modifications in controlling the degradation process, we investigate a case study using E. coli equipped with a synthetic circuit designed to modulate magnesium breakdown. This enables us to evaluate how the presence or absence of the genetic circuit influences the wireless signal behavior of the implant. Electroactive bacteria contain redox-active proteins that enable their interaction with extracellular electron acceptors, such as solid metals or metal oxides, and donors through electrical current production or consumption, respectively, as part of the extracellular electron transfer (EET) pathway[33,34]. The natural ability of microbes like *Geobacter* and *Shewanella* to exchange electrons with electrodes allows them to respire various inorganic and organic molecules for survival in redox-stratified environments. In their anaerobic respiration, these organisms transfer electrons to inorganic minerals via nanowires, from an intracellular quinol pool to inner membrane proteins, and through to the outer membrane, which likely contains a soluble redox-active shuttle for distant electrodes[33,35]. Current production occurs upon lactate catabolism, while current consumption happens during intracellular fumarate reduction[35].

While our implementation focuses on synthetic gene circuits in E. coli, the concept of microbial interaction with conductive materials is well-established in electroactive bacteria. Understanding the EET mechanisms in such microbes provides valuable context for how living cells can influence material properties and wireless behavior in biohybrid systems. Synthetic biology enables new opportunities to create bioelectronic sensors by heterologously expressing c-type cytochrome proteins, which facilitate interaction with physical or electrical components for electrochemical applications[33–43]. The MtrCAB pathway is crucial for this process, as it requires cytochrome c maturation (Ccm) proteins for the proper functioning of the system[39]. In this work, we aim to equip E. coli with the ability to perform EET, turning it into a bioelectrochemical sentinel cell capable of molecular level tracking and advancing real-time monitoring of body dynamics by sensing molecules of interest, such as disease biomarkers.

## Results

### Genetically modified bacteria

Synthetic biology tools offer us to engineer bacterial cells to create an electronic communication system between living and non-living modulus that have application areas in biosensing[34,44]. In nature, bacterial cells evolved to transfer the electrons outside the bacterial membrane to preserve their viability and reduce metal oxides found in their habitat during anaerobic respiration[34,36,44–50]. The genetic circuit design approach and protein engineering strategies can be combined to develop biological systems using the biological pathways of electrons[51]. A synthetic electron conduit can be set up via the introduction and expression of the variable essential cytochrome's protein cascade in E. coli, borrowing the necessary cytochrome encoding

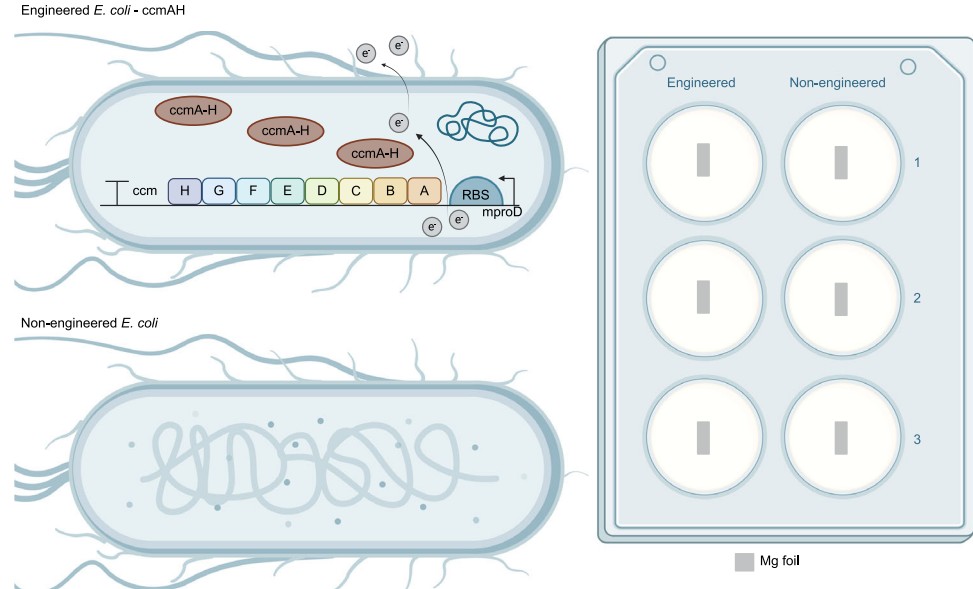

**Fig. 2 | Experimental set-up that compares the activity of CcmA–H expressing E. coli BL21 bacterial cells with non-engineered E. coli BL21.** Created in BioRender. *Bilir, A. (2025)* https://BioRender.com/z85u1w9.

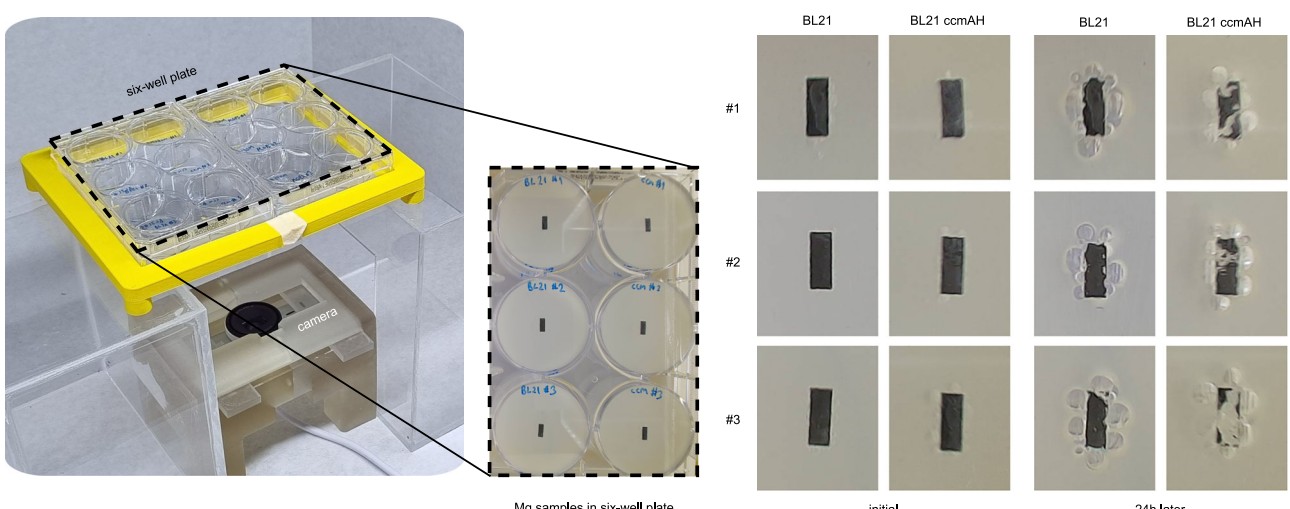

**Fig. 3 |** Experimental set-up that compares the activity of CcmA–H expressing E. coli BL21 bacterial cells with non-engineered E. coli BL21 (left). Mg foil strips of 2 × 5 mm subjected to degradation experiments with E. coli BL21 CcmA–H, which contains open gene circuits, and non-engineered E. coli BL21 cells that do not express recombinant proteins (right).

genes from *Shewanella oneidensis* and writing it to E. coli's cytochrome protein NapC, a native homolog of CymA[33], to create an electron transfer path from the cell interior to the inorganic material outside the cell[34,35,37,39–42].

Here, we envisaged that the cell-based sensor will express the required electron transfer proteins in response to detecting a molecule of interest and results in electrochemical change as an outcome would serve as a diagnostic tool. For this purpose, E. coli BL21 cell factories are engineered to express cytochrome c maturation proteins from *Shewanella oneidensis* MR-1 strain for enhancing electron flux flows in electron transfer systems[37,41]. Genetic circuits are established for the production of CcmA–H proteins under minimal constitutive promoter in order to avoid possible metabolic burden in bacterial cells. The construction of the responsible plasmid is performed via molecular cloning techniques (see Additional Supplementary file - Supplementary Data 1). After verification of the cloning by sequencing, an

experimental set-up that compares the activity of CcmA–H expressing E. coli BL21 bacterial cells with non-engineered E. coli BL21 cells is designed, as seen in Fig. 2.

**Degradation speed control**
In order to test the degradation speed control mechanism outlined, an experimental set-up shown in Fig. 3 is prepared. The set-up provides visual feedback of the biodegradation. 5 mm by 2 mm samples of 25 μm thick magnesium foil are located in the six-well plate as shown. The samples are prepared using MITS Autolab and fixed at the bottom of each well using biocompatible silicone. A camera located under the six-well plate is programmed to take a photo every three minutes. Fig. 3 shows the photos taken with a 24 h interval.

The images collected are converted to binary format and the pixel count is plotted, as seen in Fig. 4. The solid lines represent the average number of pixels, while the shaded areas indicate the standard

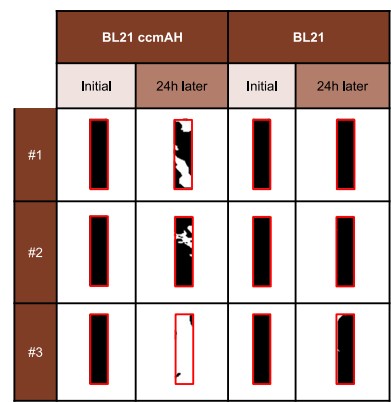

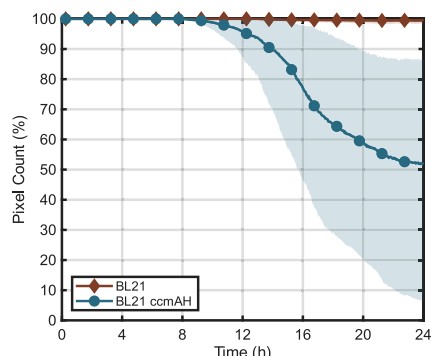

**Fig. 4 | Binary format strip images** showing the initial state and the state after 24 h of exposure to engineered E. coli BL21 CcmA–H and non-engineered E. coli BL21 (left) and pixel counts obtained from the binary-format foil images (right). The shaded area represents the standard deviation.

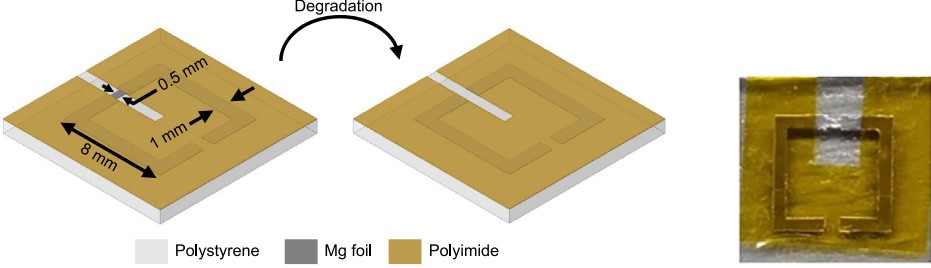

**Fig. 5 | Biodegradable passive implant antenna design** and its envisaged biodegradation steps (left), and the prototyped antenna fixed on polystyrene (right).

deviation. From this graph, it is evident that E. coli BL21 CcmA–H, which contains constitutively active gene circuits, degrades magnesium more rapidly compared to E. coli BL21 that does not express recombinant proteins. This indicates that the gene expression of E. coli BL21 CcmA–H accelerates the degradation of magnesium foil.

### Implant antenna design

The passive implant antenna is designed such that a split ring resonator is going to turn into a segmented ring resonator after the degradation, as seen in Fig. 5. The section that sits at the second split's location is designed to be 0.5 mm wide. Furthermore, the remaining sections are covered with another layer of substrate to make sure that the resonant frequency follows the predetermined pattern.

The passive implant antenna is numerically analyzed using ANSYS HFSS 2025R1 in a waveguide, as seen in Fig. 6. It is sandwiched between 3-(N-morpholino)propanesulfonic acid (MOPS) and muscle-mimicking phantom, which is an approximate representation of how the bio-hybrid implant is going to be positioned at system-level simulations. For the muscle phantom, frequency-dependent dielectric properties of muscle tissue are used[52]. The dielectric properties of MOPS are measured using DAK SPEAG 3.5 and inserted into the numerical analysis. In order to observe the resonances clearly, the conductivity values of both media are set to zero. The aim of the analysis is to estimate the resonant frequency of the structure in the media; hence, conductivity does not have a significant effect. The estimated resonant frequency of the non-degraded and degraded implant antenna is 1.16 and 1.91 GHz, respectively, as seen in Fig. 6. The range of the frequency of operation is selected to be 1–2 GHz considering the aimed implant depth and the antenna size. The optimum frequency range for in-body communications depend on both antenna size and implantation depth[53]. The designed antenna is prototyped, as seen in Fig. 5. 25 μm thick magnesium foil is plotted with MITS AutoLab and then fixed on an 11 × 11 mm polystyrene substrate of 1 mm thickness using

biocompatible silicone. Polystyrene substrate was chosen due to its transparent nature, allowing the degradation to be tracked optically to validate the wireless connection.

Figure 7 shows the biodegradation of the prototyped implant in MOPS. The optical feedback taken from the camera demonstrates that the degradation is occurring as anticipated. The split ring structure turns into a segmented ring at 23:50 hours of the degradation experiment. The moment when the transition occurs can be observed in the zoomed in images of the 0.5 mm wide section.

### On-body reader antenna design

To establish the chipless RFID link between the bio-hybrid implant and the outside world, a two-port on-body reader antenna is designed. On-body antennas are frequently used in wearable devices. Depending on the application, these antennas may establish off-body, on-body, or in-body communication links. In-body links are particularly challenging due to the reflection at the air-body interface and significant signal loss within the body. The antenna's performance is influenced by its location on the human body, as the effective permittivity varies with the predominant tissue type. To maximize the propagation into the human body and establish low coupling between the two ports of the reader antenna, a cross-slot structure is used, as seen in Fig. 8.

The on-body antenna designed is based on the antenna previously designed by the authors[54]. Figure 8 shows the simulated and measured S-parameters of the antenna when located on a muscle block. According to the measurements, it is operational between 0.8 and 2.3 GHz, where coupling between the ports is less than −30 dB.

### Phantom-based validation of wireless cell-based sensing

Figure 9 presents the detailed 3D model used for system-level simulations, along with the experimental setup and the electrical properties of both the numerical and physical phantoms. A plexiglass container is

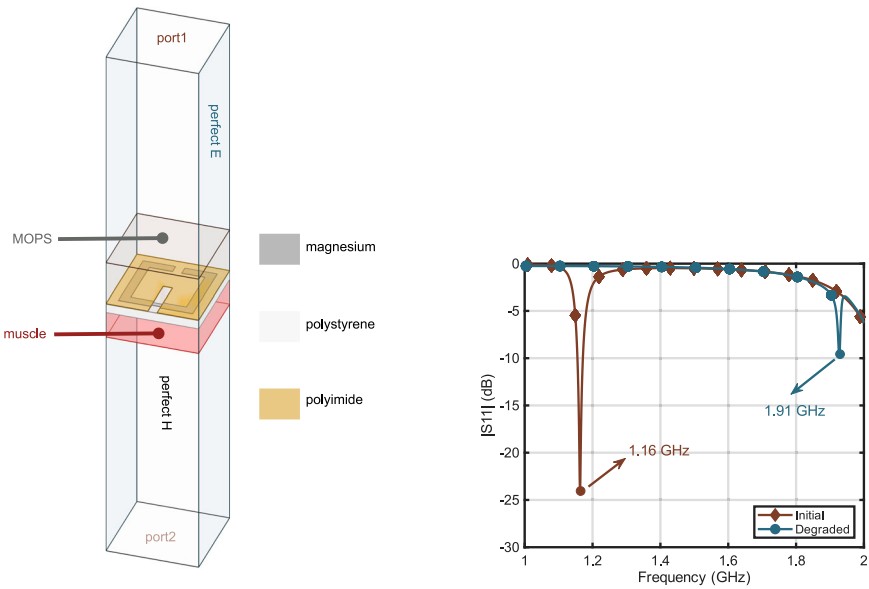

**Fig. 6 | Simulation model to determine the resonant frequency of the passive implant antenna.**

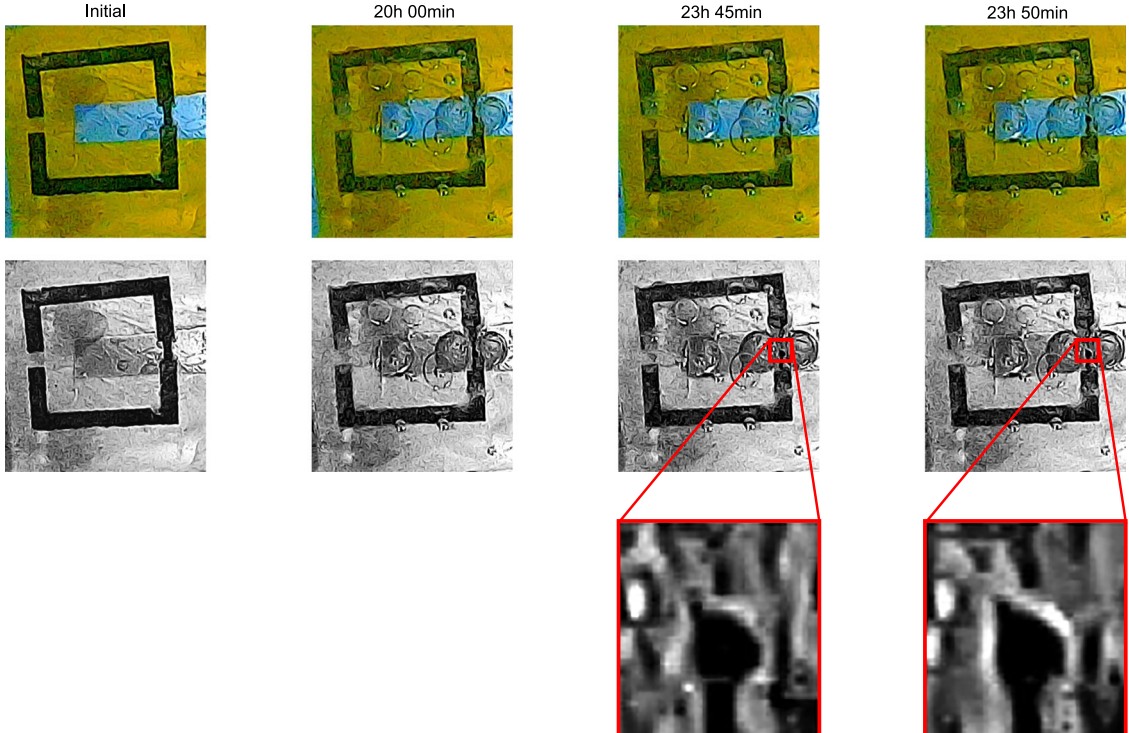

**Fig. 7 | Degradation test of the implant antenna in MOPS media to evaluate whether it degrades as expected: transition from split ring to segmented ring.**

filled with liquid muscle phantom. The bio-hybrid implant is located inside a 3D-printed cup filled with MOPS to support the bacteria. The 3D-printed cup is then submerged into the muscle phantom. The plexiglass container has a window on one of its side walls. The on-body antenna is secured in a 3D-printed holder, which is used to seal the window. The separation between the on-body antenna and the bio-hybrid implant is set to 25 mm, which represents the depth of the implant inside the human body.

The simulations were performed using ANSYS HFSS 2025R1, and the implant antenna's resonant frequency was observed at approximately 1.2 GHz. Detailed simulation results are provided in the Supplementary Information.

The biodegradation of the implant antenna is monitored both visually and electromagnetically. The ports of the on-body reader antenna are connected to the VNA (Rohde & Schwarz ZNLE6). A 4K camera (Logitech MX) is located under the container using a 3D-printed stage. The camera and the VNA are connected to a laptop which controls the measurements. Complex S-parameters are measured every 5 min over 24 h, while simultaneously the image of the implant is saved. The process is automated with a Python code. In order to minimize the effects of the environment, the whole set-up is located inside our 3 × 3 × 3 m anechoic chamber.

The muscle phantom is required to be transparent to allow visual feedback. It is developed using deionized water, glycerol, and salt. The

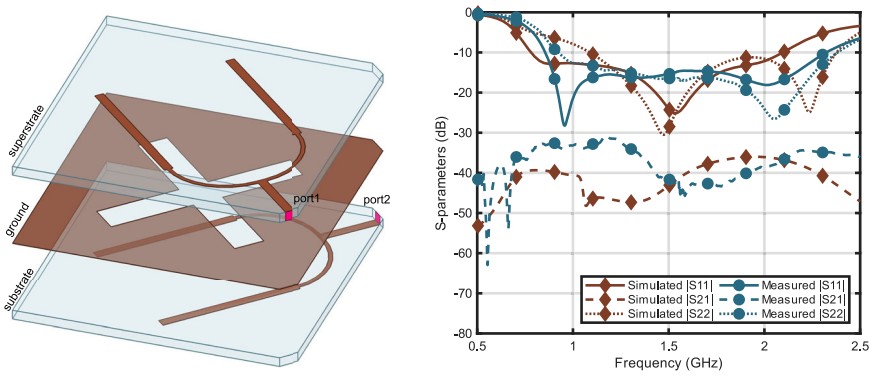

**Fig. 8 |** The on-body antenna geometry and the comparison of simulation and measurement of the magnitudes of the S-parameters of the on-body antenna when it is placed on human muscle tissue. Solid, dashed, and dotted lines represent $|S_{11}|$, $|S_{21}|$, and $|S_{22}|$, respectively.

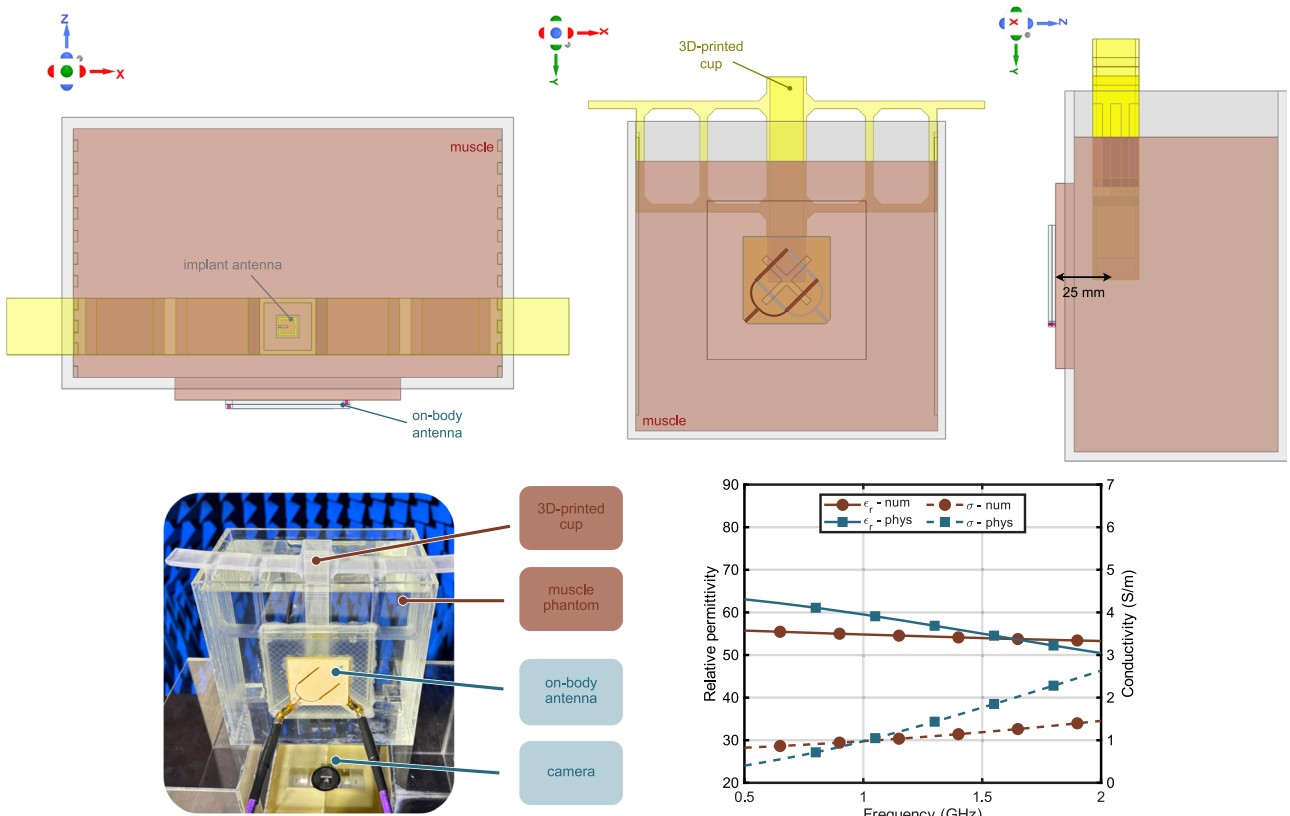

**Fig. 9 |** The simulation model from different views (top). The experimental set-up for system level tests (bottom-left) and the electrical properties of the numerical and physical phantoms where solid and dashed lines represent relative permittivity and conductivity, respectively (bottom-right).

measured dielectric properties of the phantom are compared to[52], as seen in Fig. 9. The deviation from the target dielectric constant and the conductivity values at 1.16 GHz are 6.6 and 15.9%, respectively.

In order to monitor the biodegradation electromagnetically, the transmission coefficient between the ports of the on-body reader antenna is used, as shown in Equation (1). The transmission coefficient is calibrated with the coefficient measured at the previous time point. By doing so, the change is detected, as seen in Fig. 10. It can be observed that the time at which the disconnection occurs in the visual feedback is aligned with the data collected with the on-body reader antenna.

$$\Delta S_{21}[f,n] = S_{21}[f,n] - S_{21}[f,n-1]$$
$$|\Delta S_{21}[f,n]|(dB) = 20\log_{10}|\Delta S_{21}[f,n]|$$

(1)

It is important to note that the sensing approach here is based on detection of the existence of the resonance. Hence, the sensing is immune to resonance shifts related to environmental changes in the vicinity of the implant. The exact frequency of operation is not important as long as the resonance is within the on-body reader antennas operation bandwidth.

To determine the maximum implant depth at which the implant antenna's resonant frequency remains detectable, measurements were conducted using a non-biodegradable implant antenna fabricated on an RO3003 substrate. Measurements were performed at depths ranging from 25 to 65 mm, in 10 mm increments. The results indicate that the antenna's resonance is clearly detectable up to a depth of 55 mm. Detailed measurement results are provided in the Supplementary Information.

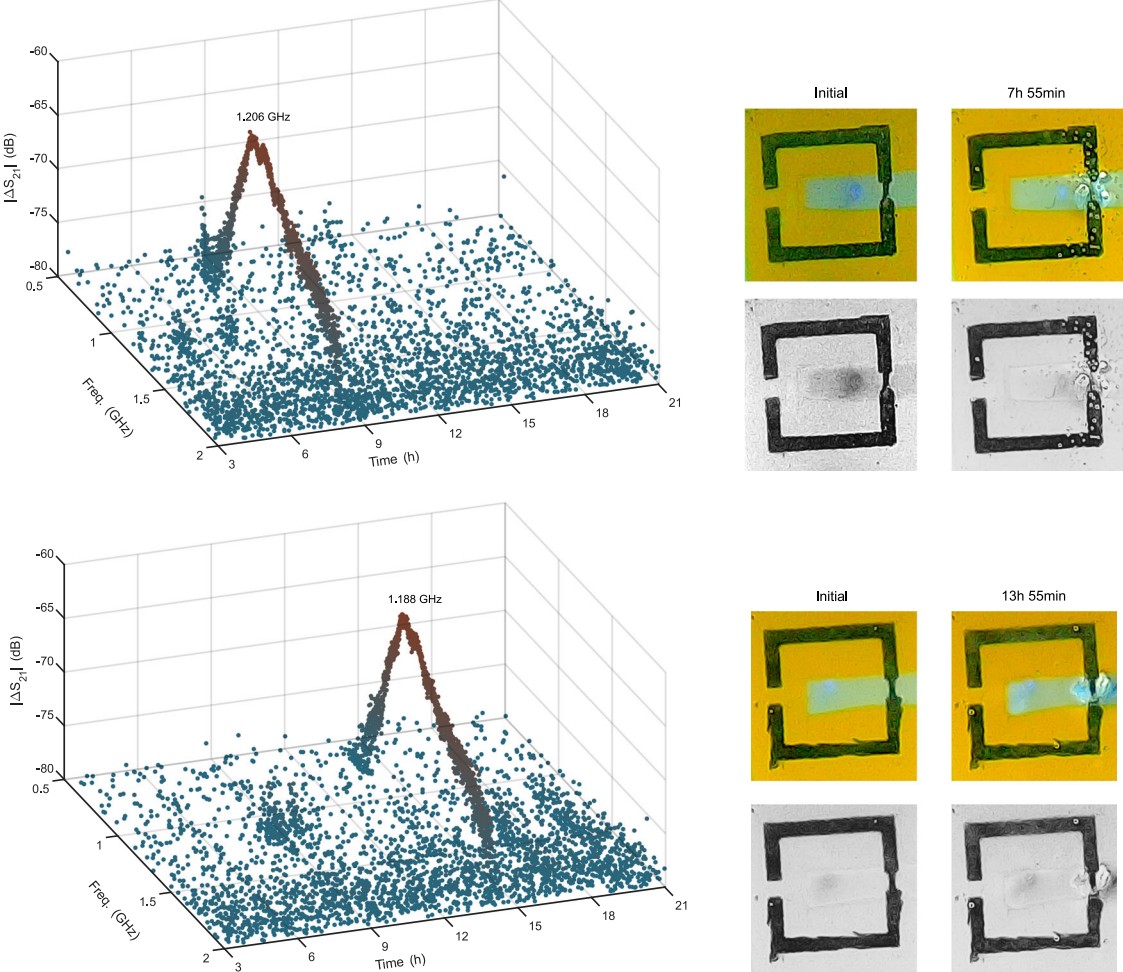

**Fig. 10 |** The calibrated $|S_{21}|$ values over time and the initial and post-degradation images when the implant antenna is exposed to engineered E. coli BL21 CcmA–H (top) and non-engineered E. coli BL21 (bottom).

## Discussion

A bio-hybrid implant composed of genetically modified bacteria and a biodegradable passive implant antenna is developed. The implant antenna is prototyped using 25 μm thick magnesium foil. The genetically modified bacteria are shown to control the biodegradation speed of the magnesium implant, where the degradation speed is envisaged to be used for sensing at the molecular level.

An on-body antenna, designed on a muscle phantom, covers the operational frequency range of the passive implant antenna as it degrades. This on-body antenna is used to wirelessly monitor the biodegradation speed of the implant via backscatter communication. Molecular-level, cell-based sensing is achieved at an implant depth of 25 mm in the muscle phantom.

In this study, we engineered E. coli bacterial machinery to express the cytochrome c (cyt c) maturation complex (CcmA–H), enabling electrical interaction with a metal surface and creating communication across the biotic-abiotic bridge. Engineering E. coli as a non-native host to harbor EET pathways broadens the application field of bioelectronics, as it allows control over transfer rates and processes for various substrates according to need[36,55,56]. The design of these bacterial systems holds promise for creating electronic sentinel cells for biomedical applications.

We engineered the E. coli BL21 (DE3) strain to express the CcmA–H maturation pathway under a constitutively active promoter to avoid metabolic overload and ensure the intended function. It is shown that BL21 CcmA–H bacterial cells degrade magnesium faster than non-engineered BL21 cells. The bacterial machinery is designed to express the cytochrome c protein complex to enhance electron flux for the purpose of controlled implant degradation.

Adapting a non-native host to perform EET functions is challenging due to the complexity of native systems, such as those in *Geobacter* and *Shewanella* species. Therefore, the essential parts of the EET pathways are selectively engineered for optimal activity[36]. To enhance bioelectronic activity, the CcmH gene can be modified through hybridization with the CcmI gene's C-terminal from *S. oneidensis* MR-1, based on findings related to EET efficiency changes with C-terminal mutations[41]. This strategy improves the compatibility of the cyt c system from *S. oneidensis* with E. coli[37,41]. However, the hybrid CcmH protein affects the inner membrane protein CymA without influencing other cyt c proteins[41].

The hybrid CcmH protein can therefore be employed in Mtr-expressing E. coli harboring the direct EET pathway[41], though a tunable induction system is needed to balance the post-translational modifications required for c-type cytochromes (mtr and ccm). This enables controlled MtrCAB expression along with improved cell growth[39]. The stoichiometry of overall cyt c in E. coli can be evaluated using ferrozine assay, redox assays, or x to assess redox activity in future experiments[34,41,57].

At the system-level, it has been demonstrated that E. coli BL21 CcmA–H degrades the implant in approximately 8 h, in contrast to non-engineered BL21, which requires approximately 14 h. This result is validated through visual feedback. This work represents the first demonstration of a wireless link between a cell-based passive sensor located inside a body mimicking phantom and the outside world.

Our results provide proof-of-concept verification for the usability and integration of engineered cellular sensors as electrically passive implantable devices, requiring no integrated circuits or batteries, but only a reflector antenna. These are wirelessly connected to the outside world. Given the immense capability of cells to detect various molecules, this approach holds potential for real-time monitoring of disease progression, prognosis, and drug effectiveness—without requiring additional invasive steps.

Genetically encoded cellular biosensors, particularly whole-cell sensors (WCS), have emerged as powerful tools in synthetic biology due to their ability to detect and transduce environmental signals with high specificity and modularity. These systems leverage engineered genetic circuits composed of standardized biological parts—promoters, transcription factors, RNA-based regulators, and other genetic elements—that can be assembled to create sophisticated sensing and response modules[58,59].

Central to these circuits are genetic logic gates, state machines which enable the implementation of complex decision-making processes within living cells. These gates perform Boolean operations (AND, OR, NOT, NAND, NOR, XOR, etc.) by integrating multiple input signals—such as regulatory proteins, peptides, small molecules, or nucleic acids—and generating a precise output response. The design of these circuits often employs well-characterized genetic parts, such as inducible promoters, transcriptional regulators, riboswitches, and RNA interference mechanisms, to achieve predictable and tunable behavior[60,61].

One particularly powerful tool in this context is the toehold switch, a synthetic RNA-based regulatory element that enables highly specific, programmable translational control. Toehold switches are engineered RNA hairpins that sequester ribosome binding sites and start codons, preventing translation in the absence of a complementary trigger RNA. Upon binding to the target trigger RNA, the hairpin unfolds, exposing the ribosome binding site and initiating translation of a downstream reporter gene. This mechanism allows for ultra-sensitive, sequence-specific detection of nucleic acid biomarkers, with tunable dynamic ranges and minimal background noise[62].

Target specificity can be finely tuned by engineering the sensor components to recognize particular biomarkers associated with disease states, environmental contaminants, or metabolic conditions. For example, a WCS can be programmed to detect a disease-specific peptide or small molecule by incorporating a responsive genetic element that binds or interacts selectively with the target. Upon recognition, the circuit can initiate a cascade of genetic events leading to a measurable output signal.

To facilitate signal transduction, these biosensors can be engineered to produce electrical or optical signals in response to target detection. For electrical signals, the circuit may induce electron flow through conductive proteins, such as cytochromes, metal-binding peptides, or engineered nanowire-like structures, resulting in detectable current changes at the cell-material interface. Alternatively, optical outputs can be generated via reporter genes encoding fluorescent proteins, luciferases, or other luminescent molecules, enabling real-time, non-invasive monitoring.

These generated signals can induce local electromagnetic changes at the cell-material interface, which can be harnessed for wireless communication, remote sensing, or integration with electronic read-out devices. Such capabilities open avenues for developing autonomous, implantable biosensing platforms that operate seamlessly within biological environments.

In addition to the specific recognition of the biomarker signals at the molecular level, biocompatibility of the overall system also holds a critical importance. Biocompatibility is an essential factor in the development of in-body living biosensing systems, as it significantly influences the safety, performance, and durability of the device within a biological context. To achieve optimal biocompatibility, we will implement surface modification strategies that concentrate on protein-based materials and functionalities derived from signaling peptides. These surface functionalization methods are intended to reduce negative immune responses and improve the system's integration with surrounding tissues.

In particular, we will use protein-based materials, including biopolymers and components of the extracellular matrix, to enhance cellular compatibility and mitigate inflammatory reactions. Moreover, we will immobilize signaling peptides onto the system's surface to promote specific cellular interactions, enhance sensor stability, and facilitate targeted signal transduction.

These strategies aim to deliver a more resilient and sustainable functionality to the biosensing system, ensuring reliable performance over prolonged periods within the intricate biological environment. By improving biocompatibility through these surface modifications, the system will be better positioned to operate effectively, minimize potential adverse effects, and ultimately provide precise, real-time monitoring of physiological parameters in vivo.

Overall, the integration of genetic logic gates and modular sensing components within WCS provides a versatile and programmable framework. This approach allows for the creation of highly specific, robust, and multifunctional biosensors tailored for applications ranging from environmental monitoring to medical diagnostics and therapeutic interventions in the realm of synthetic biology.

However, there remains significant room for improvement. To achieve a robust output signal, it is necessary to integrate additional molecular sensors and connect them to the existing cytochrome conduit. This will require further investigation using transcription factors and genetic regulatory systems.

## Methods

### Bacterial cell maintenance and genetic circuit design

For the molecular cloning experiment, the E. coli DH5α strain (purchased from New England Biolabs, USA) was utilized as it contains recA1 and endA mutations. Therefore, the insert stability is increased due to diminished recombinase activity, and DNA quality is increased, giving high transformation efficiency[63]. The protein expression experiments were performed in the E. coli BL21 (DE3) (New England Biolabs, USA) as they were deficient in proteases, giving a high level of recombinant protein expression. The constructed responsible plasmids were transformed into chemically competent cells via heat shock transformation protocol. Both bacterial strains were inoculated in Lysogeny Broth (LB) liquid medium or LB agar (Sigma Aldrich, Germany) and incubated at 30 °C or 37 °C for bacterial cell growth. The liquid cultures were incubated in a shaking incubator. Bacterial cells were stored at -80 °C (VWR, Freezer USA) in a freezing medium including 25% glycerol (Sigma Aldrich, Germany) in LB media for long-term maintenance.

For constructing the pZs mProD CcmA–H kan plasmid, the CcmA-G cassette was amplified with PCR (Biorad C1000 Thermal cycler, USA) P1 and P2 primers (purchased from Bioligo A. S., Turkey), as seen in Table S1. The pZs backbone, including the mProD promoter region, was amplified with P3 and P4 primers. The isolated PCR products were assembled via the isothermal assembly kit by following vendors protocol (Synbiotik, Turkey).

### Antenna prototyping and testing

The on-body antenna was fabricated using the LPKF ProtoMat S103 on a Rogers RO6010 substrate. Since the design consists of two separate layers, three 1 mm diameter holes were drilled into the substrates to ensure proper alignment. After aligning the layers, they were bonded together using epoxy resin.

Next, 90-degree SMA connectors were soldered to the two antenna ports. To secure the antenna onto the plexiglass container, a 3D-printed insert was used. This insert was fabricated with an SLA 3D-

printer, Anycubic Photon M3 Max, using water-washable resin. The antenna was affixed to the 3D-printed insert with epoxy resin, and the insert was then glued to the plexiglass container using the same adhesive. Epoxy resin was specifically chosen to prevent leakage from the liquid phantom. Finally, Kapton tape was applied to prevent direct contact between the microstrip on the antenna and the lossy phantom.

The implant antenna was prototyped using 99.9% Mg foil (Changsha Rich Nonferrous Metals Co., Ltd, China) with a thickness of 25 μm. The foil was shaped using the MITS Autolab PCB prototyping machine. The Mg implant antenna was then attached to an $11 \times 11$ mm polystyrene substrate using biocompatible silicone.

To set up the measurement system, a 3D-printed cup designed to contain the implant antenna, the bacteria, and the growth medium was fabricated using an SLA 3D printer with water-washable resin. The cup was equipped with fixers on both sides to secure it to the plexiglass container. Additionally, the fixing parts and the inner surfaces of the plexiglass container featured interlocking male-female strips, each 5 mm wide, to ensure that the 3D-printed cup, and consequently the implant antenna, was always positioned at a consistent distance from the on-body antenna.

To monitor the degradation of the implant antenna over time, the measurement setup also included a dedicated camera system. A 3D-printed holder was fabricated to securely mount the camera in a fixed position, ensuring a clear and unobstructed view of the implant antenna throughout the entire experiment. This setup allowed for continuous optical observation. For the same reason, the phantom, which was designed to mimic the electrical properties of human muscle tissue, was developed to be transparent. This transparency allowed both optical and electromagnetic observations to be conducted simultaneously.

### Protein expression, magnesium degradation and analysis

For protein expression experiments, the sequence-verified plasmid (plasmids were sequence verified by Genewiz, USA) was transformed into chemically competent E. coli BL21 (DE3) (New England Biolabs, USA) cells. The bacterial cells were grown overnight in LB liquid media (purchased from Sigma Aldrich, Germany) supplemented with a proper antibiotic, kanamycin (Sigma Aldrich, Germany), for the responsible cell group. The groups were specified as follows: E. coli BL21 (DE3) harboring pZs mProD CcmA−H kan, which was later depicted as BL21 CcmA−H, non-engineered E. coli BL21 (DE3) that simply indicated as BL21, and lastly the media itself. The following day, the cells were collected via centrifugation at 5000 rpm (~4000 × g) (VWR 5810 centrifuge with cooling) for 5−10 min. The cell pellets were washed with $1 \times$ MOPS (Sigma Aldrich, Germany) minimal media once, which was followed by centrifugation. The washed cell pellets were concentrated to the same OD600 value (measured with Varian Cary 5000 UV Vis spectrometer) and resuspended in a 1:5 culture volume of $1 \times$ MOPS media supplemented with 0.2% (w/v) final concentration of glucose and 0.4% (w/v) final concentration of sodium lactate besides an appropriate antibiotic if required. The additives were also provided to the media group.

Three replicates of Mg strips, placed in a 6-well plate, were incubated with the corresponding cell or media groups (3 mL each) at room temperature. Images were captured every 3 min over a 24 h period.

Each image was analyzed using MATLAB R2025a. After converting the images to binary scale, the number of black pixels was counted. All data are presented as mean ± standard deviation.

### Ethics

This study does not involve experiment involving animals, human participants, or clinical samples.

### Reporting summary

Further information on research design is available in the Nature Portfolio Reporting Summary linked to this article.

### Data availability

All data supporting the findings of this study are available within the article and its supplementary files. Any additional requests for information can be directed to, and will be fulfilled by, the corresponding authors. Source data are provided with this paper.

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

## Acknowledgements

This work was supported by the Scientific and Technological Research Institution of Turkey (TUBITAK) under project number 120C131, S.D. Also, the authors would like to thank Prof. Arda Deniz Yalcinkaya and Asst. Prof. Cansu Canbek Ozdil for insightful discussions. The authors gratefully acknowledge financial support from Synbiotik Biotechnology.

## Author contributions

S.D. proposed the project idea, supervised the work, conceptualized the experiments and acquired funding. A.B designed and fabricated the implantable resonators and the reader antenna, and carried out in vitro testing. U.O.S.S. supervised synthetic biology experiments. U.O.S.S and M.Y. designed the engineered bacteria, did molecular cloning, genetic engineering studies, and molecular characterization studies. A.B. wrote the original draft. S.D., M.Y., and U.O.S.S contributed to the writing of the original draft. All authors contributed to reviewing and editing the manuscript.

## Competing interests

The authors declare no competing interests.
