## [Transparent Peer Review file · Nature Communications]

Wireless In-body Sensing through Genetically Engineered Bacteria

Corresponding Author: Dr Sema Dumanli

Version 0:

Reviewer comments:

Reviewer #1

(Remarks to the Author)

This work introduces a novel wireless in-body sensing approach that harnesses genetically engineered E. coli to trigger controlled degradation of a passive implant antenna. By translating the bacteria's molecular response into measurable electromagnetic signals via backscatter communication, the system enables battery-free, real-time monitoring of specific molecules inside the body. This bio-hybrid concept represents a pioneering step toward integrating synthetic biology with RF sensing technologies for next-generation implantable diagnostics. My comments are below:

- Biodegradability Clarification: The authors are encouraged to elaborate on their long-term vision regarding full biodegradability of the implant. Currently, not all layers of the structure are biodegradable, which raises questions about its eventual in vivo compatibility.
- S-Parameter Completeness: While the paper presents both simulated and measured S_{11} and S_{21} results for the two-port on-body antenna, it lacks S_{22} data. Including S_{22} would provide a more complete characterization of the antenna's performance.
- Clarity of Resonant Frequency in Fig. 10: Figure 10 effectively demonstrates the sensing process through a 3D calibrated S_{21} plot over time. However, the resonant frequency is not clearly discernible. A complementary 2D plot or visual markers should be added to enhance interpretability.
- Missing Figure Reference: There appears to be a missing figure number reference at line 271, which should be corrected for clarity and consistency.

Reviewer #2

(Remarks to the Author)

This work proposes an interesting method for molecular detection using genetically engineered E. coli. The engineered bacteria alters the degradation rate of a magnesium (Mg) foil. This Mg foil functions as an implanted antenna, and its frequency response changes as it degrades. An on-body antenna is used to wirelessly detect this change. The system is passive and batteryless, which is promising for in-vivo sensing.

Below are my comments:

1. Page 1, Line 6: The statement "EM waves of which wavelength is at cellular scale cannot propagate into the lossy human tissues" is not technically accurate. While high-frequency EM waves (e.g., in the THz range) are strongly attenuated in lossy tissue, they can still propagate to a limited extent. A more accurate description would acknowledge the high attenuation rather than claiming an inability to propagate, especially under biosafety constraints.
2. Page 2, Line 45: The cited references for electrochemical and molecular sensing are outdated (mostly over 10 years old). Recent advances in molecular sensors, including in-vivo applications, should be incorporated. For example, reference [10.1109/ISSCC19947.2020.9063036] presents relevant recent work. Please expand the literature review to include more current and comprehensive studies in this area.
3. Why was the antenna designed to operate around the GHz frequency range? Is this choice driven solely by the physical dimensions of the antenna, or are there other considerations such as tissue penetration or sensitivity?

4. Please include electromagnetic simulation results (e.g., HFSS) and setup showing the coupling between the on-body and implanted antennas.
5. The phantom-based experiment uses a 25 mm separation between the on-body and implanted antennas. What is the maximum tolerable separation distance for reliable signal detection? How does the performance degrade with increased distance or tissue variation?
6. What is the concentration and spatial distribution of the bacteria? These factors may significantly influence the degradation kinetics and repeatability of the response.
7. Is it feasible to replace the benchtop vector network analyzer (VNA) with a portable or wearable system for real-time monitoring? If so, what are the sensitivity requirements of such a system?
8. What is the intended target application or implantation site for this sensor? If the implanted Mg antenna experiences deformation due to tissue pressure or implantation geometry, how would that affect its resonance and detection performance?
9. Is the sensor intended for binary (presence/absence) molecular detection, or can it support quantitative analysis based on the degradation rate or resonance shift?
10. Please include a more detailed figure showing the implanted antenna setup in the phantom experiment. The current figures lack sufficient anatomical and spatial clarity.
11. Line 199: What is the time interval between consecutive measurements? This detail is important to understand the temporal resolution of the sensor response.
12. Line 271: There appears to be a citation error at Line 271 (shown as “??”). Please correct this for clarity and completeness.

Overall Evaluation

This paper presents a promising approach to batteryless, wireless molecular sensing using engineered bacteria and degradable Mg antennas. However, to strengthen the manuscript, the technical claims should be revised for scientific accuracy, the literature review should be updated, and additional implementation details (simulations, figures, experimental parameters) should be provided.

Reviewer #3

(Remarks to the Author)

SUMMARY

This manuscript presents an innovative wireless biosensing platform that integrates genetically engineered bacteria with a passive magnesium-based implant antenna to enable in vivo molecular sensing via backscatter communication without integrated circuits or batteries.

The paper demonstrates proof-of-concept for a synthetic-biology driven approach, wherein engineered bacteria express the cytochrome c maturation complex (CcmA–H) to modulate extracellular electron transfer, thereby accelerating magnesium degradation. The magnesium degradation induces a measurable shift in antenna resonance by shifting its structure from a split ring to a segmented ring. This degradation is validated visually and electromagnetically, demonstrating the applicability of this approach. The experimental setup, which included plate reader assays and a transparent muscle phantom combined with a custom-designed on-body reader antenna, demonstrated bacteria-mediated magnesium degradation and enabled wireless signal transduction at an implant depth of 25 mm.

In the overall, this is a novel approach that could potentially be further developed into therapeutic and diagnostic devices.

MAJOR COMMENTS

1. As indicated by the authors, this approach could be implemented to detect specific molecules, physiological conditions, or diseases using biological sensors as input for such systems. However, the study does not demonstrate molecular specificity. Instead, it demonstrates the constitutive expression of the system. A specificity demonstration could make the manuscript stronger. However, given the scope of this paper and the novelty of the system, a thorough discussion of how ligand-responsive circuits could be integrated is sufficient. This discussion must include specific examples of sensors that could regulate the system.
2. A thorough discussion on the long-term stability of the bacteria/antenna implant under physiological conditions is required.
3. How would the system perform following pulse/transient inputs? It would be helpful (but not mandatory) if an inducible system could be characterized under the conditions described in Figure 3, for example, by replacing the constitutive expression with IPTG to drive such transient inputs. Would a toggle switch for the detection of pulse signals be handy? This must be discussed.

MINOR COMMENTS

1. The main text does not address Figure 2, which should probably be mentioned at the end of the second paragraph of 2.1
2. Figure legends must be revised to become more detailed and informative
3. The text detailing the ccmAH system in the upper bacteria of Figure 2 is not readable and must be enlarged.

4. Figures 1, 2, and 3 would better be combined.
5. In Figure 3, the annotations are made in handwriting and should be made similarly to Figure 4.
6. In Figure 3, it would be better to focus on the magnesium rather than on the entire well. Currently, it is hard to see the degradation.

Version 1:

Reviewer comments:

Reviewer #1

(Remarks to the Author)

Authors replied to all the raised comments.

Reviewer #2

(Remarks to the Author)

Thank you for the revision. I'm not satisfied with the claim that the sensing approach is "immune to resonance shifts" because detection "depends on the existence of the resonance." This statement conflicts with results shown in the manuscript, where changes in device geometry/material environment clearly shift the resonance. In practice, detuning also alters coupling, loaded Q, and notch depth/contrast, all of which affect detectability.

Please provide more information to clarify this question.

1. Quantify tolerance to detuning.
2. Define the detection criterion.

Is detection based on a magnitude/phase threshold, derivative, or a sweep-and-peak-pick routine? Please specify minimum detectable resonance depth (in dB) and how this threshold holds across detuning and lossier tissue.

Reviewer #3

(Remarks to the Author)

The revised version of the manuscript is stronger. The authors have fully answered my concerns. Their response to the other reviews seems satisfying. I therefore recommend accepting this manuscript for publication.

Topic: Reply to reviewers for Manuscript

We thank the editor and reviewers for their interest and accurate remarks on the manuscript entitled “Wireless In-body Sensing through Genetically Engineered Bacteria: AntennAlive” by Ahmet Bilir, Merve Yavuz, Urartu Ozgur Safak Seker, and Sema Dumanli which was submitted to Nature Communications.

We greatly appreciate the reviewers’ feedback, which tremendously helped us to improve the manuscript. Please find below the reviewers’ comments listed together with our replies. The final version of the manuscript has been revised to address the reviewer’s comments. All modifications have been colored red. Additionally, the figures have been updated to ensure that all plots and text elements are in vector format, except for Fig. S1 and Fig. S2.

REVIEWER COMMENTS

Reviewer #1 (Remarks to the Author):

This work introduces a novel wireless in-body sensing approach that harnesses genetically engineered E. coli to trigger controlled degradation of a passive implant antenna. By translating the bacteria’s molecular response into measurable electromagnetic signals via backscatter communication, the system enables battery-free, real-time monitoring of specific molecules inside the body. This bio-hybrid concept represents a pioneering step toward integrating synthetic biology with RF sensing technologies for next-generation implantable diagnostics. My comments are below:

- **Biodegradability Clarification:** The authors are encouraged to elaborate on their long-term vision regarding full biodegradability of the implant. Currently, not all layers of the structure are biodegradable, which raises questions about its eventual in vivo compatibility.

We are thankful to the author for the constructive comments about our manuscript. We understand and agree the comments on the biocompatibility concerns of the system we are presenting here. Also stated by the reviewer this is a proof-of-concept study aiming to demonstrate the biological and physical possibility of building a working biohybrid system. However, we foresee an in body real world application of the system with enhanced ligand based sensing – transmitting capabilities. As a matter of the fact such in body systems durability and functionality/ usability, heavily rely on the biocompatibility of the system. In the upcoming design we are working on using a biocompatible and

biodegradable polymer system including protein-based biopolymers. Additionally, we are planning surface functionalization of the biohybrid system with bioactive signalling peptides which can turn ON/OFF biological responses. To summarize these future perspectives, we have added the following part to the main text of the manuscript (**Line 338-354**):

“Biocompatibility is an essential factor in the development of in-body living biosensing systems, as it significantly influences the safety, performance, and durability of the device within a biological context. To achieve optimal biocompatibility, we will implement surface modification strategies that concentrate on protein-based materials and functionalities derived from signalling peptides. These surface functionalization methods are intended to reduce negative immune responses and improve the system's integration with surrounding tissues.

In particular, we will use protein-based materials, including biopolymers and components of the extracellular matrix, to enhance cellular compatibility and mitigate inflammatory reactions. Moreover, we will immobilize signalling peptides onto the system's surface to promote specific cellular interactions, enhance sensor stability, and facilitate targeted signal transduction.

These strategies aim to deliver a more resilient and sustainable functionality to the biosensing system, ensuring reliable performance over prolonged periods within the intricate biological environment. By improving biocompatibility through these surface modifications, the system will be better positioned to operate effectively, minimize potential adverse effects, and ultimately provide precise, real-time monitoring of physiological parameters in vivo.”

- **S-Parameter Completeness:** While the paper presents both simulated and measured S_{11} and S_{21} results for the two-port on-body antenna, it lacks S_{22} data. Including S_{22} would provide a more complete characterization of the antenna's performance.

The S_{22} parameter has now been included in Figure 8 to provide a more complete characterization of the antenna's performance. The caption has been updated for further clarification.

- **Clarity of Resonant Frequency in Fig. 10:** Figure 10 effectively demonstrates the sensing process through a 3D calibrated S_{21} plot over time. However, the resonant frequency is not clearly discernible. A complementary 2D plot or visual markers should be added to enhance interpretability.

The authors would like to thank the reviewer for their suggestion. Markers have been added to the plots in Figure 10 to clearly indicate the resonant frequency and improve interpretability of the sensing process.

- Missing Figure Reference: There appears to be a missing figure number reference at line 271, which should be corrected for clarity and consistency.

The missing figure reference at line 271 (line 378 in the updated manuscript) has been corrected to ensure clarity and consistency.

Reviewer #2 (Remarks to the Author):

This work proposes an interesting method for molecular detection using genetically engineered E. coli. The engineered bacteria alters the degradation rate of a magnesium (Mg) foil. This Mg foil functions as an implanted antenna, and its frequency response changes as it degrades. An on-body antenna is used to wirelessly detect this change. The system is passive and batteryless, which is promising for in-vivo sensing.

Below are my comments:

1. Page 1, Line 6: The statement “EM waves of which wavelength is at cellular scale cannot propagate into the lossy human tissues” is not technically accurate. While high-frequency EM waves (e.g., in the THz range) are strongly attenuated in lossy tissue, they can still propagate to a limited extent. A more accurate description would acknowledge the high attenuation rather than claiming an inability to propagate, especially under biosafety constraints.

The authors would like to thank the reviewer for the correction. The sentence in the abstract (Line 6) has now been revised to:

“EM waves with wavelengths at the cellular scale experience high attenuation, which limits their propagation into lossy human tissues.”

2. Page 2, Line 45: The cited references for electrochemical and molecular sensing are outdated (mostly over 10 years old). Recent advances in molecular sensors, including in-vivo applications, should be incorporated. For example, reference [10.1109/ISSCC19947.2020.9063036] presents relevant recent work. Please expand the literature review to include more current and comprehensive studies in this area.

The authors would like to thank the reviewer for their correction. The following paragraphs (line 48 – line 63) have been added:

“More recent studies have started to address this limitation by developing platforms for real-time in vivo molecular sensing. For example, Saunders et al. introduces a set of generalizable molecular switch designs based on aptamers and antibodies for

continuous in vivo detection. These sensors employ optical methods to convert molecular binding events into measurable signals, but they require a fiber-optic connection for signal acquisition and monitoring [20]. Similarly, Chen et. al. demonstrates an optical sensing system capable of detecting specific molecules in vivo when implanted in the blood vessels of freely moving rats for up to one week [21]. Moutsiopoulou et. al. also explores aptamer-based molecular sensing using optical moving techniques [22].

Chien et. al. reports an aptamer-based sensor for detecting kanamycin in freely moving rats, with data transmitted wirelessly via Bluetooth. However, this system requires battery and signal processing electronics [23].

Chen et. al. discusses an implantable aptamer-based sensor that is inductively coupled to an external reader. Although it allows for wireless power and data transmission, the implant still contains integrated electronics for signal processing [24].

References:

[20] J. Saunders, I. A. Thompson, and H. T. Soh, "Generalizable molecular switch designs for in vivo continuous biosensing," *Accounts of Chemical Research*, vol. 58, no. 5, pp. 703–713, Feb. 2025.

[21] Y. Chen et al., "A biochemical sensor with continuous extended stability in vivo," *Nature Biomedical Engineering*, May 2025.

[22] A. Moutsiopoulou, D. Broyles, E. Dikici, S. Daunert, and S. K. Deo, "Molecular aptamer beacons and their applications in sensing, imaging, and Diagnostics," *Small*, vol. 15, no. 35, Jul. 2019.

[23] J. -C. Chien, S. W. Baker, K. Gates, J. -W. Seo, A. Arbabian and H. T. Soh, "Wireless Monitoring of Small Molecules on a Freely-Moving Animal using Electrochemical Aptamer Biosensors," 2022 IEEE Biomedical Circuits and Systems Conference (BioCAS), Taipei, Taiwan, 2022, pp. 36-39.

[24] S. Chen, T.-L. Liu, Y. Dong, and J. Li, "A Wireless, regeneratable cocaine sensing scheme enabled by allosteric regulation of pH sensitive aptamers," *ACS Nano*, vol. 16, no. 12, pp. 20922–20936, Dec. 2022."

3. Why was the antenna designed to operate around the GHz frequency range? Is this choice driven solely by the physical dimensions of the antenna, or are there other considerations such as tissue penetration or sensitivity?

The antenna dimensions as well as the implant depth affects the choice of operating frequency. There exists an optimum frequency range for a set implant depth. Here the implant antenna remains detectable up to an implant depth of 55 mm (this issue will be discussed in comment 5 as well). The operating frequency in the GHz range was chosen as a trade-off between physical size and penetration capability.

If deeper implantation is required, a lower operating frequency would be preferable due to its improved penetration in biological tissue. However, this would also result in larger antenna dimensions for both the implant and the on-body antennas. Conversely, if compact size is the primary constraint, the operating frequency could be increased, but at the expense of reduced penetration depth—limiting the antenna's use to more superficial implant locations.

In order to clarify this choice, we have added the following sentences (line 176 – line 178):

“The range of the frequency of operation is selected to be 1 GHz to 2 GHz considering the aimed implant depth and the antenna size. The optimum frequency range for in-body communications depend on both antenna size and implantation depth [54].”

Reference:

[54] A. K. Skrivervik, M. Bosiljevac, and Z. Sipus, “Antennas for implants: design and limitations,” in *Bioelectromagnetics in Healthcare: Advanced Sensing and Communication Applications*, W. Whittow, Ed. Stevenage, U.K.: IET (Institution of Engineering and Technology), 2022, pp. 81–90.

4. Please include electromagnetic simulation results (e.g., HFSS) and setup showing the coupling between the on-body and implanted antennas.

In the manuscript, the simulation setup has now been included at the top of Figure 9 in the manuscript.

The following sentence has been added at line 204:

“Figure 9 presents the detailed 3D model used for system-level simulations, along with the experimental setup and the electrical properties of both the numerical and physical phantoms.”

To further clarify the simulation procedure, the following sentence has been added at line 213:

“The simulations are performed using ANSYS HFSS, and the implant antenna's resonant frequency is observed at approximately 1.2 GHz. Detailed simulation results are provided in the Supplementary Information.”

In addition, the electromagnetic simulation results have been added to the Supplementary Information as Figure S3. The calibrated simulation exhibits behavior consistent with the measurement results, with the resonant frequency of the implant antenna observed around 1.2 GHz. Furthermore, the following paragraph has been added to the Supplementary Information under the section: “EM Simulation Results” to describe the simulation methodology and findings:

“The simulation model is shown in Figure 9 of the manuscript. Simulations were performed both with and without the implant antenna, and the transmission coefficient was calibrated using the method described in the manuscript. The resulting data are presented in Figure S3. Consistent with the measurement results shown in Figure 10, the implant antenna exhibits a resonant frequency near 1.2 GHz.”

5. The phantom-based experiment uses a 25 mm separation between the on-body and implanted antennas. What is the maximum tolerable separation distance for reliable signal detection? How does the performance degrade with increased distance or tissue variation?

To investigate the maximum tolerable separation between the on-body and implant antenna, a non-biodegradable implant antenna was fabricated on an RO3003 substrate to facilitate repeated measurements. Note that the resonance frequency of this non-biodegradable antenna slightly deviates from that of the biodegradable version due to differences in substrate properties.

Accordingly, the following text has been added to the manuscript at line 240:

“To determine the maximum implant depth at which the implant antenna's resonant frequency remains detectable, measurements were conducted using a non-biodegradable implant antenna fabricated on an RO3003 substrate. Measurements were performed at depths ranging from 25 mm to 65 mm, in 10 mm increments. The results indicate that the antenna's resonance is clearly detectable up to a depth of 55 mm. Detailed measurement results are provided in the Supplementary Information.”

Figure S4 in the Supplementary Information has been updated to show the MOPS-mimicking phantom used to replicate the electrical properties of MOPS, along with the frequency-dependent electrical properties of deionized water. Additionally, measurement results for various implant depths have been added in Figure S6. These results confirm that the implant antenna remains detectable up to a depth of 55 mm.

To evaluate the effect of tissue conductivity, Figure S7 compares the calibrated transmission coefficients when the implant antenna is placed in a high-conductivity MOPS-mimicking phantom versus a low-conductivity deionized water environment. The results show that the implant antenna exhibits a higher quality factor in the low-

conductivity medium, resulting in a sharper resonance. It is also important to note that a decrease in the medium's permittivity cause the resonant frequency of the implant antenna to shift higher. This also demonstrates the effect of tissue variation on the sensing.

The following paragraphs have been added to the Supplementary Information under the section: "EM Measurement Results" to detail the measurement setup and results:

"Measurements were conducted using the same setup shown in Figure 9. To replicate the electrical properties of MOPS, a MOPS-mimicking phantom was prepared using a salt-water solution, as illustrated in Figure S4. Additionally, a water-glycerol mixture was used to mimic muscle tissue, and the effect of glycerol concentration on the mixture's electrical properties is shown in Figure S5.

To investigate the reliable sensing depth, an implant antenna is prototyped on an RO3003 substrate with a thickness of 1.57 mm, as shown in Figure S6 (a).

The measurement procedure involved three steps: first, the S-parameters were recorded without the implant present in the setup ($S_{xy,1}$). Next, the implant antenna was placed inside a 3D-printed cup, and a second set of S-parameters was recorded ($S_{xy,2}$). Finally, the antenna was removed, and a third set of S-parameters was measured ($S_{xy,3}$). This procedure was repeated for implant depths ranging from 25 mm to 65 mm, in 10 mm increments.

For each measurement, the same calibration method described in the manuscript was applied. The initial measurement ($S_{21,1}$) served as the reference for calibration. The calibrated transmission coefficient in the final measurement without the antenna, defined as $|\Delta S_{21,3}| = |S_{21,3} - S_{21,1}|$, represents the noise level, since both cases lacked the implant. The calibrated transmission coefficient when the implant was present, $|\Delta S_{21,2}| = |S_{21,2} - S_{21,1}|$, reveals the resonant frequency of the implant antenna.

The results are shown in Figure S6 (b-f). As the implant depth increases, the backscattered signal from the implant antenna becomes weaker. At 55 mm depth, the signal is still discernible, while at 65 mm, it is no longer visible.

Note that since the antenna is non-biodegradable and fabricated on a commercial substrate to simplify the measurement process, its resonant frequency differs slightly from that of the biodegradable version.

To demonstrate the effect of tissue conductivity on the resonance quality, the calibrated transmission coefficients are presented in Figure S7, where the implant depth is fixed at 35 mm. The measurement was repeated using deionized water instead of the MOPS-mimicking phantom. As shown in Figure S4, deionized water has significantly lower conductivity compared to the MOPS-mimicking phantom. Due to the reduced

conductivity of the surrounding medium, the implant antenna exhibits a higher quality factor, resulting in a sharper and more pronounced resonance.”

6. What is the concentration and spatial distribution of the bacteria? These factors may significantly influence the degradation kinetics and repeatability of the response.

The absorbance of both non-engineered and engineered bacteria was adjusted to approximately 1.6 at 600 nm to ensure similar concentrations in both cases (Line 399). Spatial distribution has not been investigated yet. The authors agree that this is worth investigating. For the time being, the effect of all variation is included in the repeatability analysis. It can be deduced from Fig. 4 that how the degradation occurs varies from sample to sample. However, the degradation times of the non-engineered and engineered bacteria differ significantly, rendering the effect of variation negligible for this density.

7. Is it feasible to replace the benchtop vector network analyzer (VNA) with a portable or wearable system for real-time monitoring? If so, what are the sensitivity requirements of such a system?

In our current setup, we measure the complex transmission coefficient using a benchtop VNA, and then apply a calibration procedure as described in the manuscript. The measured S_{21} magnitude is approximately -40 dB. After calibration, the signal level is around -60 dB when the implant antenna is present, and around -75 dB when it is absent. The difference in the calibrated transmission coefficient (ΔS_{21}) due to the presence of the implant serves as the sensing metric.

To replace the benchtop VNA with a portable or wearable system, the critical requirement is the system’s ability to detect and resolve this differential signal level, approximately 15 dB dynamic range between the implant and no-implant cases, while maintaining sufficient phase and magnitude accuracy. In particular, a sensitivity better than -70 dB in transmission would be necessary to reliably distinguish the presence of the implant antenna, considering signal attenuation in tissue and possible environmental noise. Finally, the frequency resolution is not restrictive since our sensing does not rely on frequency shift. We detect the existence of the resonance of the implant which is an amplitude shift at a wide frequency band.

Overall, it is feasible to envision a portable or wearable system for real-time monitoring, provided that the system can achieve a sensitivity on the order of -70 dB and resolve signal differences of around 10–15 dB in a lossy environment.

8. What is the intended target application or implantation site for this sensor? If the implanted Mg antenna experiences deformation due to tissue pressure or implantation geometry, how would that affect its resonance and detection performance?

The intended application of this sensor system is to be integrated into the tissues and organs. The implant can be deformed or the immediate tissue properties can change which will affect the resonance frequency of the implant. However, our sensing approach is immune to these kind of resonance shifts since the detection depends on the existence of the resonance. The exact frequency of operation is not important as long as the resonance is within the on-body reader antennas operation bandwidth.

In order to clarify this point, the following paragraph is added to Line 235:

“It is important to note that, the sensing approach here is based on detection of the existence of the resonance. Hence the sensing is immune to resonance shifts related to environmental changes in the vicinity of the implant. The exact frequency of operation is not important as long as the resonance is within the on-body reader antennas operation bandwidth.”

9. Is the sensor intended for binary (presence/absence) molecular detection, or can it support quantitative analysis based on the degradation rate or resonance shift?

At the moment the sensor is envisaged to be binary. We believe, for plenty of cases inside the human body, the interest is to know if the molecule of interest is above a certain threshold. We are aiming for those applications.

10. Please include a more detailed figure showing the implanted antenna setup in the phantom experiment. The current figures lack sufficient anatomical and spatial clarity.

To address the reviewer’s comment, a detailed 3D model illustrating the exact location of the implanted antenna within the phantom setup has been added to the manuscript at the top of Figure 9. This figure provides improved anatomical and spatial clarity.

11.Line 199: What is the time interval between consecutive measurements? This detail is important to understand the temporal resolution of the sensor response.

The time interval between consecutive measurement are 5 minutes. It is given at Line 221 in the updated manuscript.

12. Line 271: There appears to be a citation error at Line 271 (shown as “??”). Please correct this for clarity and completeness.

The missing figure reference at line 271 (line 378 in the updated manuscript) has been corrected to ensure clarity and consistency.

Overall Evaluation

This paper presents a promising approach to batteryless, wireless molecular sensing using engineered bacteria and degradable Mg antennas. However, to strengthen the manuscript, the technical claims should be revised for scientific accuracy, the literature review should be updated, and additional implementation details (simulations, figures, experimental parameters) should be provided.

We would like to thank the reviewer for their insightful comments which made our manuscript stronger.

Reviewer #3 (Remarks to the Author):

SUMMARY

This manuscript presents an innovative wireless biosensing platform that integrates genetically engineered bacteria with a passive magnesium-based implant antenna to enable in vivo molecular sensing via backscatter communication without integrated circuits or batteries. The paper demonstrates proof-of-concept for a synthetic-biology driven approach, wherein engineered bacteria express the cytochrome c maturation complex (CcmA–H) to modulate extracellular electron transfer, thereby accelerating magnesium degradation. The magnesium degradation induces a measurable shift in antenna resonance by shifting its structure from a split ring to a segmented ring. This degradation is validated visually and electromagnetically, demonstrating the applicability of this approach. The experimental setup, which included plate reader assays and a transparent muscle phantom combined with a custom-designed on-body reader antenna, demonstrated bacteria-mediated magnesium degradation and enabled wireless signal transduction at an implant depth of 25 mm.

In the overall, this is a novel approach that could potentially be further developed into therapeutic and diagnostic devices.

MAJOR COMMENTS

1. As indicated by the authors, this approach could be implemented to detect specific molecules, physiological conditions, or diseases using biological sensors as input for such systems. However, the study does not demonstrate molecular specificity. Instead, it demonstrates the constitutive expression of the system. A specificity demonstration could make the manuscript stronger. However, given the scope of this paper and the novelty of the system, a thorough discussion of how ligand-responsive circuits could be integrated is sufficient. This discussion must include specific examples of sensors that could regulate the system.

We thank the reviewer for their constructive comments. We agree that the system can be extended to respond to specific molecular disease signals. In this study, given its proof-of-concept nature, we employed a constitutively active system. However, the ultimate goal of this work is to develop a living antenna system capable of responding selectively upon encountering a disease-related ligand.

In response to the reviewer's point, we have added the following paragraphs to the main text of the manuscript:

“Genetically encoded cellular biosensors, particularly whole-cell sensors (WCS), have emerged as powerful tools in synthetic biology due to their ability to detect and transduce environmental signals with high specificity and modularity. These systems leverage engineered genetic circuits composed of standardized biological parts—promoters, transcription factors, RNA-based regulators, and other genetic elements—that can be assembled to create sophisticated sensing and response modules [58], [59].

Central to these circuits are genetic logic gates, state machines which enable the implementation of complex decision-making processes within living cells. These gates perform Boolean operations (AND, OR, NOT, NAND, NOR, XOR, etc.) by integrating multiple input signals—such as regulatory proteins, peptides, small molecules, or nucleic acids—and generating a precise output response. The design of these circuits often employs well-characterized genetic parts, such as inducible promoters, transcriptional regulators, riboswitches, and RNA interference mechanisms, to achieve predictable and tunable behavior [60], [61].

One particularly powerful tool in this context is the toehold switch, a synthetic RNA-based regulatory element that enables highly specific, programmable translational control. Toehold switches are engineered RNA hairpins that sequester ribosome binding sites and start codons, preventing translation in the absence of a complementary trigger RNA. Upon binding to the target trigger RNA, the hairpin unfolds, exposing the ribosome binding site and initiating translation of a downstream reporter gene. This mechanism

allows for ultra-sensitive, sequence-specific detection of nucleic acid biomarkers, with tunable dynamic ranges and minimal background noise [62].

Target specificity can be finely tuned by engineering the sensor components to recognize particular biomarkers associated with disease states, environmental contaminants, or metabolic conditions. For example, a WCS can be programmed to detect a disease-specific peptide or small molecule by incorporating a responsive genetic element that binds or interacts selectively with the target. Upon recognition, the circuit can initiate a cascade of genetic events leading to a measurable output signal.

To facilitate signal transduction, these biosensors can be engineered to produce electrical or optical response to target detection. For electrical response, the circuit may induce electron flow through conductive proteins such as cytochromes, metal-binding peptides, or engineered nanowire-like structures. Alternatively, optical outputs can be generated via reporter genes encoding fluorescent proteins, luciferases, or other luminescent molecules, enabling real-time, non-invasive monitoring.

The electrical response can induce structural changes in the material that interfaces the cells. These changes can be harnessed for wireless sensing with electronic readout devices. Such capabilities open avenues for developing autonomous, implantable biosensing platforms that operate seamlessly within biological environments.

Overall, the integration of genetic logic gates and modular sensing components within WCS provides a versatile and programmable framework. This approach allows for the creation of highly specific, robust, and multifunctional biosensors tailored for applications ranging from environmental monitoring to medical diagnostics and therapeutic interventions in the realm of synthetic biology. “

Reference:

[58] B. Saltepe, E. U. Bozkurt, M. A. Gungen, A. E. Cicek, and U. O. Seker, “Genetic circuits combined with machine learning provides fast responding living sensors,” *Biosensors and Bioelectronics*, vol. 178, p. 113028, Apr. 2021.594

[59] I. C. Koksaldi et al., “SARS-COV-2 detection with de novo-designed synthetic riboregulators,” *Analytical Chemistry*, vol. 93, no. 28, pp. 9719–9727, Jun. 2021.

[60] D. Akboga, B. Saltepe, E. U. Bozkurt, and U. O. Seker, “A recombinase-based genetic circuit for Heavy Metal Monitoring,” *Biosensors*, vol. 12, no. 2, p. 122, Feb. 2022.

[61] R. E. Ahan, B. M. Kirpat, B. Saltepe, and U. O. Seker, “A self-actuated cellular protein delivery machine,” *ACS Synthetic Biology*, vol. 8, no. 4, pp. 686–696, Feb. 2019.

[62] I. C. Koksaldi et al., “RNA-based sensor systems for affordable diagnostics in the age of Pandemics,” *ACS Synthetic Biology*, vol. 13, no. 4, pp. 1026–1037, Apr. 2024.

2. A thorough discussion on the long-term stability of the bacteria/antenna implant under physiological conditions is required.

We would like to thank the reviewer for their comment. There are two aspects to the long-term stability of the implant. From the material point of view, the implant is envisaged to be short term as it is fully biodegradable. Magnesium has been shown to be a safe material for implantation¹.

The second aspect is the sensing stability. In time, the implant can be deformed or the immediate tissue properties can change which will affect the resonance frequency of the implant. However, our sensing approach is immune to these kind of resonance shifts since the detection depends on the existence of the resonance. The exact frequency of operation is not important as long as the resonance is within the on-body reader antennas operation bandwidth.

In order to clarify this point, the following paragraph is added to Line 235:

“It is important to note that, the sensing approach here is based on detection of the existence of the resonance. Hence the sensing is immune to resonance shifts related to environmental changes in the vicinity of the implant. The exact frequency of operation is not important as long as the resonance is within the on-body reader antennas operation bandwidth.”

3. How would the system perform following pulse/transient inputs? It would be helpful (but not mandatory) if an inducible system could be characterized under the conditions described in Figure 3, for example, by replacing the constitutive expression with IPTG to drive such transient inputs. Would a toggle switch for the detection of pulse signals be handy? This must be discussed.

We thank the reviewer and agree with the point raised. We have discussed this extensively in the section we added to the text in response to the reviewer's first comment.

MINOR COMMENTS

1. The main text does not address Figure 2, which should probably be mentioned at the end of the second paragraph of 2.1.

The authors would like to thank the reviewer for their correction. Figure 2 has been mentioned at line 144.

2. Figure legends must be revised to become more detailed and informative.

The legends for the simulated and measured S-parameter plots in Figure 8, as well as the frequency-dependent electrical properties of the numerical and physical phantoms in Figure 9, have been updated.

3. The text detailing the ccmAH system in the upper bacteria of Figure 2 is not readable and must be enlarged.

The authors would like to thank the reviewer for the comment. The text detailing the ccmAH system in the upper part of Figure 2 has been enlarged to improve readability.

4. Figures 1, 2, and 3 would better be combined.

Although we understand the reviewer's concern, considering the complicated nature of these figures, we opt for keeping the figures separate for the sake of simplicity.

5. In Figure 3, the annotations are made in handwriting and should be made similarly to Figure 4.

The annotations in Figure 3 have been made similarly to Figure 4.

6. In Figure 3, it would be better to focus on the magnesium rather than on the entire well. Currently, it is hard to see the degradation.

Figure 3 has been regenerated to focus on the magnesium strips.

¹R. Zeng, W. Dietzel, F. Witte, N. Hort, and C. Blawert, "Progress and challenge for magnesium alloys as biomaterials," *Advanced Engineering Materials*, vol. 10, no. 8, Jul. 2008.

Thank you for the revision. I'm not satisfied with the claim that the sensing approach is "immune to resonance shifts" because detection "depends on the existence of the resonance." This statement conflicts with results shown in the manuscript, where changes in device geometry/material environment clearly shift the resonance. In practice, detuning also alters coupling, loaded Q, and notch depth/contrast, all of which affect detectability.

Please provide more information to clarify this question.

1. Quantify tolerance to detuning.
2. Define the detection criterion.

Is detection based on a magnitude/phase threshold, derivative, or a sweep-and-peak-pick routine? Please specify minimum detectable resonance depth (in dB) and how this threshold holds across detuning and lossier tissue.

Thank you for raising this important point. We agree that our earlier statement ("immune to resonance shifts") was ambiguous, and we have clarified it in the revised manuscript. Detection indeed relies on the existence of a resonance, and therefore it is necessary to quantify the degree of detuning that can be tolerated and to clearly define the detection criterion. The following information is included in the Supplementary Information.

1. Tolerance to Detuning

Detuning of the implant resonance may occur due to changes in the permittivity of the surrounding medium. Since the implant antenna is directly exposed to the bacterial medium, its effective permittivity is initially close to that of water (≈ 80), as shown in Figure S4 (left). To quantify the effect of detuning, we performed additional waveguide simulations by varying the relative permittivity of the surrounding medium from 80 down to 20, as seen in Figure R1.

- At $\epsilon_r = 80$ (MOPS-like medium), the resonance appears around 1.15 GHz.
- As ϵ_r decreases to 20, the resonance shifts upward to 1.75 GHz.

Importantly, in all cases the resonance remains within the operating band of the on-body reader antenna (1–2 GHz). This indicates that even significant environmental variations (e.g., due to changes in water/fat content) do not push the resonance out of the operational range. Moreover, such permittivity changes typically occur gradually. Therefore, as long as consecutive measurements are taken within sufficiently short intervals, resonance evolution due to environmental variations will appear smooth. In contrast, degradation of the implant (i.e., structural discontinuity) produces a sudden

and sharp change in the spectral response (see Fig. 10 of the manuscript), which can be unambiguously detected.

Figure R1: The effect of the permittivity of the media on the resonant frequency of the implant.

It is also worth noting that the permittivity range $\epsilon_r = 20\text{--}80$ encompasses most biological tissues and fluids in the 1–2 GHz range, excluding primarily bone and fat (reference: IT'IS database, <https://itis.swiss/virtual-population/tissue-properties/database/dielectric-properties>). Hence, the implant remains trackable in physiologically relevant environments.

Note that the maximum sensing depth will be affected by the permittivity and conductivity changes of the immediate tissue.

2. Detection Criterion

The detection process is based on analyzing the calibrated S_{21} (ΔS_{21}) response of the reader–implant link across 1–2 GHz. The following procedure is employed:

1. At each time instant, the maximum $|\Delta S_{21}|$ within the band is identified.
2. In the intact (non-degraded) state, the spectral maximum remains at the noise floor.
3. Degradation is detected once the maximum exceeds a threshold of 10 dB relative to the baseline noise level.

This criterion ensures reliable identification of sudden resonance emergence associated with implant degradation.

3. Effect of Lossy Media

The quality of the resonance depends on medium conductivity. We compared two representative cases:

- Deionized water ($\sigma \approx 0.31$ S/m at 1.2 GHz)
- MOPS phantom ($\sigma \approx 1.99$ S/m at 1.2 GHz)

Results are presented in Supplementary Figure S7 and reproduced as Figure R2 here. As expected, the resonance is sharper and more distinct in low-conductivity media (deionized water). In the lossy MOPS medium, the resonance is shallower, yet still detectable. Notably, the conductivity of MOPS is higher than most tissues and biological fluids, except cerebrospinal fluid, small intestine, and urine at this frequency (IT'IS database). Thus, our detection approach remains robust across a wide range of realistic biological conditions.

In summary, while resonance shifts do occur with detuning, our results show that:

- Resonances remain within the operational band (1–2 GHz) for $\epsilon_r = 20$ –80.
- Detection is based on a magnitude-threshold method (≥ 10 dB deviation).
- The approach remains functional in lossy media, including conditions more challenging than most tissues.

Figure R2: The effect of the conductivity of the media on the resonance quality of the implant.